# Supercritical Fluid and Conventional Extractions of High Value-Added Compounds from Pomegranate Peels Waste: Production, Quantification and Antimicrobial Activity of Bioactive Constituents

**DOI:** 10.3390/plants11070928

**Published:** 2022-03-30

**Authors:** Kaja Kupnik, Maja Leitgeb, Mateja Primožič, Vesna Postružnik, Petra Kotnik, Nika Kučuk, Željko Knez, Maša Knez Marevci

**Affiliations:** 1Laboratory for Separation Processes and Product Design, Faculty of Chemistry and Chemical Engineering, University of Maribor, Smetanova ulica 17, 2000 Maribor, Slovenia; kaja.kupnik@um.si (K.K.); maja.leitgeb@um.si (M.L.); mateja.primozic@um.si (M.P.); vesna.postruznik@um.si (V.P.); petra.kotnik@um.si (P.K.); nika.kucuk@um.si (N.K.); zeljko.knez@um.si (Ž.K.); 2Faculty of Mechanical Engineering, University of Maribor, Smetanova ulica 17, 2000 Maribor, Slovenia; 3Faculty of Medicine, University of Maribor, Taborska ulica 8, 2000 Maribor, Slovenia

**Keywords:** antimicrobial activity, antioxidants, bioactive compounds, extraction, LC-MS/MS, phenolics, phytochemistry, pomegranate, *Punica Granatum* L., secondary metabolites

## Abstract

This study is focused on different extractions (Cold Maceration (CM), Ultrasonic Extraction (UE), Soxhlet Extraction (SE) and Supercritical Fluid Extraction (SFE)) of bioactive compounds from pomegranate (*Punica Granatum* L.) fruit peels using methanol, ethanol, and acetone as solvents in conventional extractions and changing operating pressure (10, 15, 20, 25 MPa) in SFE, respectively. The extraction yields, total phenols (TP) and proanthocyanidins (PAC) contents, and antioxidant activity of different extracts are revealed. TP and PAC recovered by extracts ranged from 24.22 to 42.92 mg gallic acid equivalents (GAE)/g and 2.01 to 5.82 mg PAC/g, respectively. The antioxidant activity of extracts ranged from 84.70% to 94.35%. The phenolic compound identification and quantification in selective extracts was done using the LC-MS/MS method. The contents of different flavonoids and phenolic acids have been determined. SFE extract, obtained at 20 MPa, contained the highest content (11,561.84 μg/g) of analyzed total polyphenols, with predominant ellagic acid (7492.53 μg/g). For the first time, Microbial Growth Inhibition Rates (MGIRs) were determined at five different concentrations of pomegranate SFE extract against seven microorganisms. Minimal Inhibitory Concentration (MIC_90_) was determined as 2.7 mg/mL of SFE pomegranate peel extract in the case of five different Gram-negative and Gram-positive bacteria.

## 1. Introduction

Large amounts of waste biomass are produced on an annual basis during the production, processing and consumption of agricultural products. Among them there are also various fruit wastes (e.g., peels, seeds, shells, leaves, and pomace), which originate from the manufacture of various products, such as fresh cut fruit, canned and dehydrated fruit, juices, jams and others. The consumption and use of fruit waste for the production of value-added compounds or products with high economic value holds a great prospective [1]. In addition to nutrients present in fruits, such as minerals and vitamins, there are a number of plant-derived components—phytochemicals (e.g., carotenoids, enzymes, fiber, phytosterols, polyphenols)—that could promote health. Polyphenolic compounds (e.g., flavonoids, tannins, phenolic acids and anthocyanins) are one of the most representative bioactive compounds in fruit waste. The presence of important constituents in fruit peels, which exhibit anti-inflammatory, antioxidant, and antimicrobial effects, should be exploited as an opportunity to use these valuable raw materials as a basis for new active substances (e.g., antimicrobials), thus providing an economical and environmentally friendly way to reduce agricultural waste.

Due to the growing knowledge and awareness of pomegranate fruit (*Punica Granatum* L.) health-promoting benefits, it is increasingly being researched and used for consumption, both in its fresh form and in various products such as juices and diverse food supplements [2]. On the other hand, about 40–50% of the weight of the fruit is inedible (i.e., peels) and is discarded when consuming or producing various products from the fruit. The approximate estimate of the biological waste of pomegranate is about 9 tons per 1 ton of produced juice, consisting of approx. 78% of the peels and 22% of the seeds [3]. Due to extensive research in recent years, the chemical composition of both peels and seeds is already well known [4]. However, it should be noted that pomegranate peels (PP) are considered a rich source of polyphenols. Polyphenols present in PP include anthocyanins (i.e., cyanidin, delphinidin, pelargonidin), flavonoids (i.e., catechin, epicatechin, quercetin, rutin, kaempferol, luteolin, naringenin, hesperidin), phenolic acids (i.e., caffeic, chlorogenic, ellagic acid, and gallic acid), and ellagitannins (i.e., punicalagin and punicalin) [5,6].

The yield and content of bioactive compounds are strongly influenced by the choice of solvent and method of isolation or extraction. A detailed review article by El-Shamy and Farag [7] on novel trends in extraction methods of bioactive substances from pomegranate biowastes was recently published. According to the literature, due to the polarity of phenolics from PP, water or its hydroalcoholic mixtures, methanol, ethanol, n-hexane and acetone are effective for the extraction of bioactives from PP. The advantages of using supercritical fluids for the extraction of high value-added compounds from natural sources are well known and include solvent-free products, low operating temperature, short extraction time, selective extraction of components or the fractionation of total extracts, high extract quality, simplicity, and environmental friendliness [8]. SFE on the other hand preserves the natural properties of the obtained bioactive compounds [9]. It has recently especially been used for the production of phytochemicals from food waste and by-products [10]. While most studies focused on conventional organic solvent extractions, a very small number of studies using SFE for the extraction of bioactive compounds from PP have been found in the literature. Moreover, SFE is a remarkably promising technique for obtaining bioactives from plant material, as it is a green process that results in the recovery of relatively clean and pure extracts that are useful even for nutraceuticals, pharmaceutical products and for functional food [11].

This investigation is focused on bioactive compounds derived from *Punica Granatum* L. peels and highlights the feasibility of sustainable extraction methods (UE and SFE). It is believed that these findings could be a useful tool for the pomegranate juices industry to apply an effective and economically viable extraction process, transforming a by-product to a high added value functional product with a certain pharmacological effect. Our study is the first to provide a comprehensive insight into the impact of various extraction methods, solvents and process conditions on the recovery of high value-added compounds from PP. The importance of our study should therefore be emphasized as it involves the precise determination and comparison of the content of certain polyphenols (e.g., ellagic acid, gallic acid, epicatechin, catechin, and others) in PP extracts obtained with different solvents (methanol (MeOH), ethanol (EtOH), and acetone (Ace)) by conventional extractions and at different operating pressures (10, 15, 20, 25 MPa) by SFE, which has not yet been published. The efficiency (i.e., extraction yield) of all extraction procedures has been compared. Furthermore, the content of total phenols (TP) and proanthocyanidins (PAC) in PP extracts, their antioxidant activity and the presence of certain phenolic components were determined. The presence of several polyphenolic components (Figure 1) (e.g., ellagic and gallic acid, punicalagin, and punicalin) in PP extracts has been associated with their antibacterial activity [12].

It is known that different polyphenolic components can have a synergistic effect on inhibiting the growth of microorganisms, as they act as anti-infectives by forming complexes with proteins (interaction with sulfhydryl groups of extracellular and soluble proteins) present in the microbial cell walls and causing their lysis [12,13]. Therefore, the antimicrobial activity of SFE PP extract, which, according to preliminary analyses, contained the highest content of analyzed polyphenols, was tested. On the other hand, to the best of our knowledge, this is the first extensive study of the antimicrobial activity of SFE PP extracts, which also includes the accurate quantitative determination of the MGIRs at different concentrations of various Gram-negative, Gram-positive bacteria, and fungi. Antimicrobial activity was qualitatively determined against three species of Gram-negative bacteria, four species of Gram-positive bacteria and eight species of fungi. Furthermore, the antimicrobial efficacy of PP extract obtained by SFE was quantified by the determination of MGIR at five different concentrations and MIC_90_ values for PP extract against seven species of microorganisms.

## 2. Results and Discussion

### 2.1. Effect of Extraction Methods and Changing Solvent/Pressure on the Extraction Yield

First, a study of four different extraction methods (UE, SE, CM, and SFE) on the extraction efficiency (% *w*/*w*) was performed. The results are presented in Figure 2 (for UE, SE and CM) and Figure 3 (for SFE). When comparing the yields of the conventional extraction methods, SE provided the highest yields, followed by UE and then CM. It is also necessary to point out the influence of different extraction solvents, as the yields were the highest at all extraction methods when using MeOH as a solvent. The use of EtOH and Ace as solvents resulted in at least three times lower yields than the use of MeOH in the case of all conventional extraction methods. Thus, the highest yield (38.89%) was achieved with SE and the use of MeOH.

All observations are in line with other studies, reporting that MeOH usually produces a higher total extract yield than other solvents. For example, recently [14] the CM technique was used and different solvents were tested for the extraction of phenolics from PP. MeOH exhibited the highest yield (29.16%), followed by EtOH, Ace, chloroform, ethyl acetate, and water. It was also reported [15] that MeOH was the most efficient solvent for the extraction of PP (approx. 46.5% yield), while with the use of EtOH, the extraction yield was almost 2.5 times lower (approx. 17.7%), which is comparable with our results. 

Additionally, the Wilcoxon–Mann–Whitney test showed a significant difference (*W* = 30, *p* = 0.007) in extraction yields between MeOH and other solvents. Median extraction yield with MeOH was 27% (24.9%, 32.9%) compared to 8.02% (5.59%, 10.3%) obtained with other solvents. 

Furthermore, SFE is an emerging green method, considered to be an alternative technique for the extraction of high value bioactive compounds from natural products [16]. Therefore, it is interesting to look at the yields of our SFE of PP, using SC CO_2_ and EtOH as a co-solvent, under different operating pressures (10–25 MPa).

The highest SFE yield (12.34%) was achieved at the pressure of 10 MPa and it decreased with increasing operating pressure. However, the yields of SFE, especially those obtained at 10 and 15 MPa, are comparable to the yields of some conventional extractions (in the case of using EtOH and Ace as solvents), which is very promising.

According to the reviewed literature, a small number of studies have reported on the extraction of bioactive components from PP using SFE. Mushtaq et al. [17] used enzyme-assisted SFE for the extraction of phenolic antioxidants from PP, while Bustamante et al. [18] and Rivas et al. [19] used supercritical CO_2_ using a Box–Behnken design, where temperature, pressure and co-solvent or time were independent variables for the optimization of the process. Studies revealed that the highest yields were obtained after 2.5 h extraction at 40–50 °C and 20–30 MPa using 20% of the co-solvent. Rivas et al. [19] reported 1–1.5% yield for SFE of PP, using SC CO_2_ at 25–30 MPa and 45–55 °C, while Ara and Raofie [20] achieved 1.18% yield using SC CO_2_ and MeOH as a modifier at 35 MPa and 55 °C. Hence, the choice of EtOH as a co-solvent in our study proved to be very good, as the yields using SC CO_2_ without a co-solvent were much lower. The explanation for higher yields is in increasing the polarity of the solvent with the addition of EtOH, which is necessary because of the low polarity of CO_2_ and for better solubility and the extraction of more polar bioactive compounds (e.g., polyphenols). As a result, SFE using SC CO_2_ and EtOH as a co-solvent achieved fairly high yields of PP extracts (4.85–12.34%). Hereafter, the yields themselves could be further increased by optimizing some important variables and by using additional modern techniques in conjunction with SFE, such as enzyme-assisted SFE, which can double the extraction yield [17].

### 2.2. Content of Total Phenols, Proanthocyanidins and Antioxidant Activity of PP Extracts

The method of extraction and the use of different solvents can result in divergent contents of bioactives in the PP extracts. Therefore, a study of the content of total phenols (TP) and proanthocyanidins (PAC) in all obtained PP extracts was performed. The antioxidant potentials of the extracts were also determined using the DPPH assay. The results of the experimental work are shown in Table 1 (for UE, SE and CM) and in Figure 4 (for SFE).

As a basic, total phenols were measured using the Folin–Ciocalteu’s reagent in each extract and the results were expressed in mg GAE per gram of extract. CM and SE generally extracted higher TP contents (approx. 40 mg GAE/g), while UE and SFE resulted in lower values of TP (approx. 25 mg GA/g). The choice of solvents in the conventional extractions (UE, SE, CM) did not have such an effect on the TP content. However, it should be noted that the highest TP content in all conventional extractions was achieved using MeOH, followed by EtOH and Ace, which coincides with the polarity of the solvents (MeOH > EtOH > Ace) and literature [21,22].

It was reported [23] that MeOH has a higher capacity for extracting phenolic compounds from dried PP as EtOH and that is a better solvent for extraction of TP from PP. The TP content in the extracts is difficult to compare with other studies, as the extraction methods and the expression of the results alone (catechin equivalents, gallic acid equivalents, tannic acid equivalents, etc.) are different. However, for better perception, another study [24] showed that the phenolic contents of MeOH, EtOH, and Ace extracts by CM were found to be 78.92, 20.39, and 3.47 mg GAE/g, respectively. Further, Fawole et al. [25] reported up to 295.5 mg GAE per g of dry MeOH extract by UE, and Živković et al. [26] reported about UE with EtOH for extraction of polyphenolics from PP, which resulted in 81.61–190.94 mg GAE/g dw. Ali and Kumar [27] subjected PP to UE and SE using MeOH. SE resulted the TP content from 1.82–4.00 mg GAE/g, while UE resulted in TP content of 2.45–4.49 mg GAE/g.

As far as SFE is concerned, the operating pressure did not have a significant effect on the TP content of the samples, as all PP extracts contained approx. 24 mg GAE/g. The results are comparable to UE, while SE and CM resulted in higher TP contents. For comparison, according to the literature, a maximum of 0.8 g of tannic acid equivalents/100 g dw [28], 43.62 mg GAE/g extract [19] and 1.01–8.94 mg GAE/g [18] of TP were extracted with SFE.

Next, the PAC content was generally highest in CM extracts. The highest value (5.82 mg PAC/g) was shown by the CM Ace extract, followed by CM EtOH and then CM MeOH with 5.37 and 4.88 mg PAC/g of extract, respectively. In the case of UE and SE, however, the highest PAC content was present in the case of MeOH as a solvent with 3.71 and 4.47 mg PAC/g, respectively, while EtOH and Ace extracted lower PAC content (2.47–3.54 mg/g). These results can also be compared with SFE, which at most operating pressures (10, 20 and 25 MPa) resulted in approximately 3 mg PAC per gram of extract. For comparison, another study [15] determined approx. 3% content in EtOH and Ace extracts and 1% PAC in MeOH extract, but using catechin as a standard, while Benslimane et al. [29] reported higher PAC contents in MeOH, EtOH, and Ace PP extracts, with values of 115, 145, and 220 mg equivalent of catechin per gram of dry weight. Both studies used maceration as an extraction method, and found that Ace extracts contained the most PAC, which is also consistent with results for our conventional extractions.

Additionally, the Kruskal–Wallis test was performed to confirm statistical differences between Phenolic content of extracts prepared by different extraction methods (χ = 10.659, *p* = 0.014). CM and SE had higher phenolic content than other methods. The median of CM was 40.9 (40.5, 41.9) mg GAE/g while the median of the SE was 40.3 (39.8, 40.4) mg GAE/g. The median phenolic content of extracts obtained with SFE was 24.3 (24.3, 24.4), while the median of extracts prepared with UE was 25.4 (25.0, 25.7). A two sample *t*-test showed a significant difference (*t*(5) = 6.4429, *p* = 0.001) in the PAC content of extracts prepared using CM compared to other extraction methods. Mean PAC content of extracts prepared with CM was 5.36 (±0.47) mg PAC/g compared to 3.01 (±0.70) mg PAC/g.

However, it should be noted that the results of the studies may differ mainly due to divergent factors (e.g., variety types, climatic growing conditions, fruit ripeness, storage conditions, sample preparation, extraction method etc.) that may affect the content of bioactives and their properties [30].

The antioxidant potential of the extracts was assessed using the DPPH method. The extracts obtained via conventional extraction methods showed remarkable antioxidant activities, with more than 90% inhibition. The highest antioxidant activity (91.92–94.35%) was achieved by MeOH PP extracts (CM > SE > UE) obtained with all conventional extractions. This coincides with the fact that TPs are responsible for antioxidant activity, as MeOH extracts also contained the highest values of TP (CM > SE > UE). On the other hand, extracts obtained with SFE showed 84.7–89.5% antioxidant activity, which makes sense, as they also had a lower TP content than conventional extracts. To the best of our knowledge, there is no published study comparable to the results of the antioxidant activity of SFE PP extracts described in this study. Regarding conventional extractions, Kumar and Neeraj [31] realized that for extracts where freeze-dried PP of two varieties were used (CM), radical scavenging activity was the highest in case of MeOH and EtOH as solvents, followed by water, Ace, and hexane. All solvents except hexane gave extracts with the achieved antioxidant activity of above 90% inhibition. Furthermore, Kupnik et al. [30] determined 90.05% inhibition of EtOH PP extract obtained by SE, while Sharayei et al. [32] performed UE for extracting effective compounds from PP. Samples exhibited 15.85–88.76% antioxidative activity, using DPPH assay.

### 2.3. Content of Certain Flavonoids and Phenolic Acids in PP Extracts

Table 2 (UE, SE, CM) and Table 3 (SFE) show the proportions of the selected polyphenol components contents in analyzed PP extracts. The content of four flavonoids (catechin, epicatechin, hesperidin/neohesperidin, rutin) and four phenolic acids (caffeic acid, chlorogenic acid, ellagic acid, gallic acid) was determined. The common fact in all tables is that ellagic acid (EA) predominated in all analyzed extracts.

When comparing the presence of valuable components in conventional extracts, ellagic acid content was the highest (5601.08–6883.41 μg/g) in Ace extracts (UE > SE > CM), while gallic acid content (the second most abundant component; 1018.71–4256.04 μg/g) was the highest in the UE MeOH extract. Caffeic and chlorogenic acid were present in lower concentrations of up to 28.44 and 8.41 μg/g, respectively. Regarding flavonoids, epicatechin (350.74–1594.62 μg/g) was predominant, followed by catechin, rutin and hesperidin/neohesperidin. Overall, the highest total content of analyzed polyphenols was contained in SE Ace extract (11477.11 μg/g), followed by UE Ace, UE MeOH, CM Ace, and UE EtOH extracts of PP.

Furthermore, the content of phenolic acids and flavonoids in SFE extracts was completely comparable to conventional extracts. Ellagic acid also predominated in the case of SFE extractions, followed by gallic acid, epicatechin, catechin, rutin, hesperidin/neohesperidin and caffeic and chlorogenic acid in the lowest concentrations. Here, it is necessary to point out the SFE extract obtained at an operating pressure of 20 MPa (see Figure 5 for chromatogram), in which the highest content (11561.84 μg/g) of analyzed polyphenols was obtained, even higher than in conventional extracts. The same extract contained at least a 1.5 times higher concentration (7492.53 μg/g) of ellagic acid than other SFE extracts. The concentrations of the remaining analyzed components were higher in PP extract obtained at 20 MPa, with the exception of catechin, which was present in higher concentrations in SFE extracts obtained at 10, 15, and 25 MPa.

Many studies have also already examined the contents of polyphenolic components in various PP extracts [33,34,35], but studies about flavonoids and phenolic acids contents in SFE extracts obtained at different operating pressures have not yet been described in the literature, which makes our study a very important contribution to this field. As the pressure increases, the density of the solvent increases, which is an important factor in improving the recovery of polyphenols. The increased solvent density improves the interaction between the molecules of the solvent SC CO_2_, the co-solvent EtOH and the polyphenol content. The consequent increase in density elevates the diffusivity and solubility [36].

Studies of the solubility of the individual components in SC CO_2_ may be helpful here for explanation. For example, Putra et al. [37] found out that the solubility of epicatechin increases with increasing pressure up to 20 MPa. On the other hand, as the pressure increases from 20 to 30 MPa, the epicatechin content reduces, which is attributed to the lower selective power of SC CO_2_. Catechin also has a better and higher solubility at a pressure of 10 MPa than at 30 MPa. The results of our study are completely in line with the mentioned results, but it is necessary to take into account that the trend of solubility of individual components differs and that the change in operating pressure does not have the same effect on the recovery of different components.

### 2.4. Antimicrobial Activity of SFE PP Extract

Based on the high content of polyphenol components and the lack of information regarding antimicrobial activity of SFE extracts in the literature, the antimicrobial activity of SFE extract obtained at 20 MPa was further characterized.

Preliminarily, the antimicrobial activity of SFE PP extract was checked by disc diffusion method (DDM) against three species of Gram-negative bacteria, four species of Gram-positive bacteria, and eight species of fungi. Additionally, the microbial growth inhibition rate (MGIR) on selected bacteria and fungi was determined by broth microdilution method (BMM) at five different SFE PP extract concentrations.

The results of qualitative DDM are presented in Table 4. The zone of inhibition (in mm) is a circular area around the spot of the inhibitor in which the microorganism colonies do not grow and is thus used to measure the susceptibility of microorganisms to inhibitors.

SFE PP extract inhibited the growth of all tested Gram-negative bacteria. *P. fluorescens* (37 ± 2 mm) was the most susceptible to the addition of the extract as an inhibitor, followed by *P. aeruginosa* and *E. coli*. Furthermore, the extract was extremely effective in inhibiting the growth of Gram-positive bacteria *B. cereus* (36 ± 2 mm) as well as *S. aureus* and *S. pyogenes*, while *S. platensis* was not susceptible to the addition of the extract. SFE PP extract also inhibited the growth of some fungi of different genera, that is, *A. flavus*, *C. albicans*, *P. cyclopium*, and *T. viride*.

Based on the reviewed literature, we did not find a comparable study covering the antimicrobial activity of SFE PP extracts. However, for ease of illustration, a recent study by Kupnik et al. [30] demonstrated the antibacterial activity of EtOH PP extract (Soxhlet extraction) and H_2_O PP extract (Homogenizer-assisted extraction). EtOH extract of lyophilized PP was found to inhibit the growth of Gram-negative bacteria with the inhibition zones in the range 17–23 ± 2 mm and the growth of Gram-positive bacteria with 11–23 ± 2 mm. On the other hand, H_2_O extract of lyophilized PP was found to inhibit the growth of Gram-negative bacteria with 16–21 ± 2 mm inhibition zones and Gram-positive bacteria with 12–23 ± 2 mm inhibition zones. Hence, it should be noted that the SFE extract in our study showed larger inhibition zones at the same initial concentration of microorganisms, especially in the case of *P. aeruginosa*, *P. fluorescens* and *B. cereus*.

Given the promising qualitative results with DDM, BMM was further used for more accurate and quantitative antimicrobial activity study at different concentrations of applied SFE PP extract. A comparison of the antimicrobial activity and quantitative determination of MGIRs of SFE PP extract on the growth of Gram-negative, Gram-positive bacteria, and fungi is presented in Figure 6.

In the present study, SFE PP extract proved to be an effective inhibitor of Gram-negative bacteria. At the highest added concentration of extract (2.7 mg/mL), the growth of all Gram-negative bacteria was inhibited, resulting in 95.24 ± 3.14%, 98.10 ± 2.59%, 99.05 ± 3.09% MGIR for *E. coli*, *P. fluorescens*, and *P. aeruginosa*, respectively. With the addition of lower extract concentrations, *E. coli* and *P. fluorescens* were not susceptible, while the addition of 0.07 mg/mL SFE PP extract resulted in 38.52 ± 1.01% MGIR on *P. aeruginosa* growth. Since the highest added concentrations reached MGIR above 90%, a concentration of 2.7 mg/mL can be determined as MIC_90_ for the tested Gram-negative bacteria.

Furthermore, SFE PP extract significantly inhibited the growth of *B. cereus* (97.78 ± 3.43% MGIR) and *S. aureus* (98.46 ± 2.69% MGIR) among Gram-positive bacteria with the addition of 2.7 mg/mL (MIC_90_). Otherwise, the same concentration of SFE PP extract inhibited the growth of *S. pyogenes* with 60.07 ± 2.46% MGIR.

Regarding the inhibition of the fungus *C. albicans,* even the highest concentration applied inhibited its growth with only 42.03 ± 1.68% MGIR. However, it should be noted that the fungus was similarly susceptible to extremely lower concentrations of the extract, reaching 40.13 ± 1.64% MGIR at a nine times lower concentration (0.3 mg/mL) and 28 ± 1.38% MGIR at an almost 40 times lower concentration (0.07 mg/mL). Given the trend in other microorganisms, we anticipate that a slightly higher concentration than the tested concentrations of SFE PP extract could completely inhibit the growth of *C. albicans*, but further studies are needed for accurate data.

There are no comparable results found in the literature for the quantitative determination of MGIR for SFE PP extracts, but the results are completely comparable with EtOH and H_2_O lyophilized PP extracts from Kupnik et al. [30]. The only difference is that lower concentrations of EtOH and H_2_O extracts achieved higher MGIRs in the case of Gram-positive and Gram-negative bacteria, with the exception of H_2_O extract in the case of *P. fluorescens*, which inhibited the growth of this bacterium only at the highest concentration (2.7 mg/mL) of the extract. Therefore, our study is a very important contribution for antimicrobial activity determination of PP extracts obtained by SFE, which is considered a “green” method for the extraction of high value bioactives.

## 3. Materials and Methods

### 3.1. Materials

Acetonitrile, agar, 2,2-diphenyl-1-picrylhydrazyl (DPPH, ≥97.0%), hydrochloric acid (HCl, 37%), dimethyl sulfoxide (DMSO), ethanol (EtOH, ≥99.5%), ferrous sulfate heptahydrate (Fe(SO_4_) × 7H_2_O), Folin–Ciocalteu’s reagent (FC), gallic acid (GA, 97.5–102.5%), methanol (MeOH), n-butanol (≥99.5%), sodium carbonate (Na_2_CO_3_, ≥99.9%), peptone from soybean and yeast extract, were purchased from Sigma Aldrich^®^ (Darmstadt, Germany). Meat extract, meat peptone and sodium chloride (NaCl) were purchased from Merck (Darmstadt, Germany), Mueller–Hinton broth and potato dextrose agar were purchased from Biolife (Milano, Italy), carbon dioxide (CO_2_, purity 2.5) was purchased from Messer (MG-Ruše, Slovenia), methanol (MeOH, CHROMASOLV™ LC-MS, ≥99.9%) from Riedel-de-Haën™, Honeywell (Charlotte, NC, USA), D-(+)-glucose anhydrous was supplied from Kemika (Zagreb, Croatia), while malt extract, potato dextrose broth, and tryptic soy broth were supplied from Fluka (Buchs, Switzerland).

All microorganisms (*A. brasiliensis* (DSM 1988), *A. flavus* (DSM 818), *A. fumigatus* (DSM 819), *A. niger* (DSM 821), *B. cereus* (DSM 345), *C. albicans* (DSM 1386), *E. coli* (DSM 498), *P. aeruginosa* (DSM 1128), *P. fluorescens* (DSM 289), *S. aureus* (DSM 346), *S. cerevisiae* (DSM 1848), *S. platensis* (DSM 40041), *S. pyogenes* (DSM 11728)) were obtained from a German Collection of Microorganisms and Cell Cultures GmbH (Berlin, Germany), except *P. cyclopium* and *T. viride,* which were obtained from the Department of Agricultural Chemical Technology, Budapest University of Technology and Economics, Hungary.

Lyophilized pomegranate (*P. granatum*) peels (PP) were purchased from Alfred Galke GmbH (Bad Grund, Germany). Afterwards, the lyophilized peels were used for extractions after grinding.

### 3.2. Extractions

Extractions of lyophilized PP were performed using conventional extraction methods (UE, SE, CM) with different solvents (MeOH EtOH, and Ace). Twenty to twenty-five grams of material were put into the flask and 150 mL of solvent was added. The standard procedures for UE, SE and CM have been described in our previous publication [38]. SFE with CO_2_ and EtOH as a co-solvent was performed at different operating pressures (10–25 MPa). The supercritical fluid extraction (SFE) was performed using a semi-continuous apparatus previously presented and described by Talmaciu et al. [39]. Table 5 depicts the operating conditions for extractions.

After extractions, the solvents (MeOH, EtOH, Ace)/co-solvents (EtOH) were evaporated under reduced pressure at 40 °C, using the rotary evaporator (Büchi Rotavapor R-114, Flawil, Switzerland). The final extracts were stored and kept in a freezer at −20 °C until further use.

### 3.3. Extraction Yield Determination

The extraction yields (*η*) were expressed as the ratio between the mass of obtained extract and the initial mass of lyophilized PP. The extractions were repeated twice, and results are presented as mean values.

### 3.4. Determination of Total Phenols and Proanthocyanidins

The content of total phenols (TP) and proanthocyanidins (PAC) were determined by spectrophotometric protocols (Varian—CARY 50 UV–VIS Spectrophotometer) [40]. The TP were determined using Folin–Ciocalteu’s reagent based on colorimetric reduction/oxidation reaction and the results were reported as mg gallic acid equivalents (GAE)/g extract. Besides, the PAC were determined based on acid hydrolysis and color formation using Fe sulphate solution and results were expressed as mg PAC/g extract. All experiments were conducted in triplicates and results are reported as a mean ± standard deviation (SD).

### 3.5. Determination of Antioxidant Activity

The DPPH method was used for determining the antioxidant activity of samples [41]. In brief, extract solution prepared in MeOH was added to the 0.025 g/L MeOH solution of DPPH. The absorbance was measured at 517 nm after 15 min of incubation in the dark. Negative control was prepared by mixing MeOH with an MeOH solution of DPPH. The antioxidant activity of extracts was reported as a percentage of inhibition (I) relative to the reference solution [42]. All experiments were conducted in triplicates and results are reported as a mean ± SD.

### 3.6. Liquid Chromatography-Mass Spectrometry Analysis

To identify and quantify selected phenolic compounds, LC-MS/MS analyses were conducted according to the method previously described [41]. For analysis, the Agilent 1200 HPLC was used in tandem with Agilent 6460 QQQ JetStream ionization. The analytical column and chromatographic separation conditions were the same as in the aforementioned reference. In brief, the chromatographic column used was Agilent C18 Eclipse Plus, 150 × 4.6 mm i.d., 1.8 μm particle size. Identification of compounds in peel extracts was performed using the LC-MS/MS method in scan mode using negative polarization and the data for analytes identification and quantification are listed in Table 6.

### 3.7. Determination of Antimicrobial Activity

#### 3.7.1. Disc Diffusion Method

For qualitative analysis of the antimicrobial activity of the analyzed sample disc, the diffusion method (DDM) was used [30,43]. The concentration of microorganisms was 1–5 × 10^6^ CFU/mL. For negative control, 5% DMSO and deionized H_2_O were used. As positive controls, amoxicillin, nystatin and vancomycin (30 μg per disc) were used. All experiments were conducted in triplicates and results are reported as a mean ± SD in mm of inhibition zone.

#### 3.7.2. Broth Microdilution Method

For quantitative analysis of the antimicrobial activity of the analyzed sample, the broth microdilution method (BMM) was used [30,42,43]. Mueller–Hinton broth was used as a universal nutrient medium for all tested microorganisms. The concentration of microorganisms was 1–5 × 10^6^ CFU/mL. For negative control, 5% DMSO was used. Microbial growth inhibition rates (MGIRs) were determined at five different concentrations of added sample (2.7, 0.3, 0.2, 0.15, and 0.07 mg sample/mL of microorganism suspension). Furthermore, MIC_90_ were determined (concentrations where the sample inhibited the growth of the microorganism with at least 90% MGIR). All experiments were conducted in triplicate and results are reported as a mean ± SD.

### 3.8. Statistical Analysis

Statistical data analysis was performed to examine the differences between extraction methods and used solvents. The distribution of extraction yields, antioxidant activity, TP, PAC and selected phenolic compounds was verified using the Shapiro–Wilk statistical test. Based on the data distribution, a parametrical *t*-test was selected for normally distributed data and non-parametric Wilcoxon–Mann–Whitney and Kruskal–Wallis tests were selected for non-normally distributed data. Numerical variables are described by mean (± SD) and in the case of abnormal distribution, by median (interquartile range).

## 4. Conclusions

The present study has demonstrated that various bioactive compounds present in pomegranate peels can be released using all of the thirteen extractions (UE, SE, CM with three different solvents, and SFE at four different operating pressures) tested. It was found that the choice of solvent does not have such an effect on the content of TP and PAC, but the extraction method is an important choice. The highest content of TP and PAC was present in extracts obtained by SE and CM, and these extracts also showed the best antioxidant potential.

Conventional extraction methods are long and time consuming, while SFE is faster and is also effective for extracting high value components. In our study, it was found that SFE can be compared to conventional methods for the extraction of high value-added components. The extraction yield decreased with increasing operating pressure, which means that at a lower pressure more components present in PP were soluble in SC CO_2_ and co-solvent EtOH. However, it should be noted that, with higher operating pressure, the density of SC CO_2_ and EtOH as a modifier increased, as well as the solvent capacity and dissolution of PP bioactive compounds into the solute being enhanced. Solubility of the desired bioactive compounds increased and, therefore, a higher content of polyphenols such as phenolic acids and flavonoids was detected. The PP extract obtained by SFE at 20 MPa resulted in the highest content of analyzed polyphenolic components (11,561.84 μg/g) among all extracts, including the highest content of ellagic acid, which is well known for its bioactivities. The mentioned extract also effectively inhibited the growth of Gram-negative, Gram-positive bacteria and fungi.

This study provided valuable information regarding different extraction methods of pomegranate peels. Above all, the results of the antimicrobial study of the SFE PP extract can help fill the current gap in knowledge in the field of the precise quantitative antimicrobial efficacy of pomegranate peel SFE extracts, which can, based on the current findings, serve as an accessible and extremely potent source for the production of value-added products.

## Figures and Tables

**Figure 1 plants-11-00928-f001:**
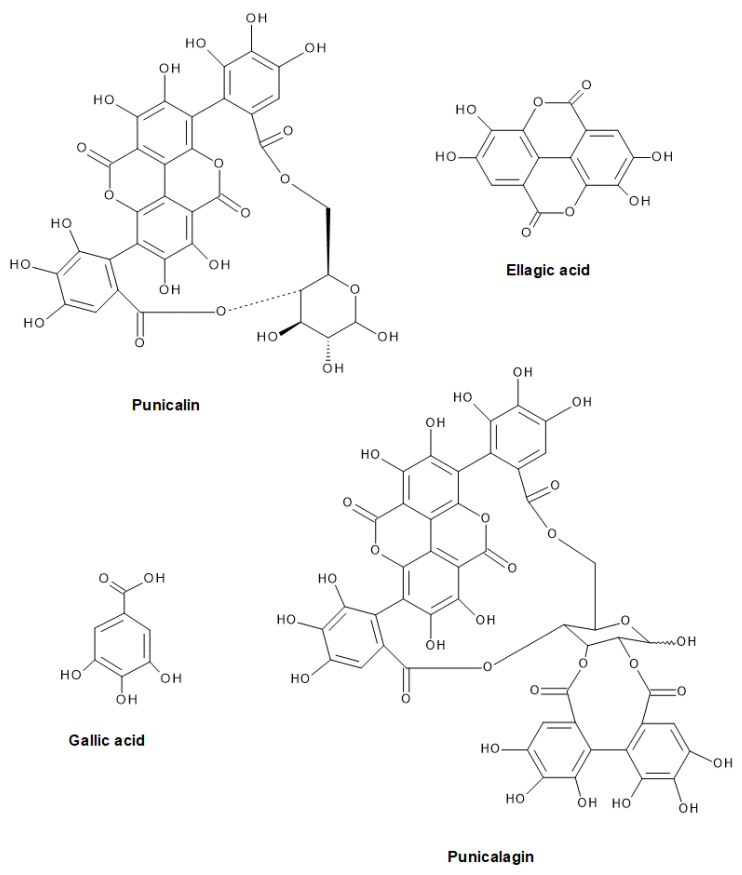
Polyphenolic compounds associated with antibacterial activity of PP.

**Figure 2 plants-11-00928-f002:**
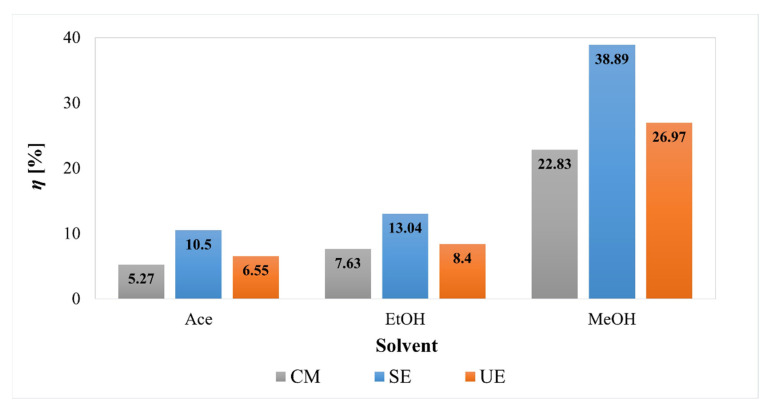
Extraction yields (*η*) obtained by cold maceration (CM), Soxhlet extraction (SE), and ultrasonic extraction (UE) of lyophilized PP.

**Figure 3 plants-11-00928-f003:**
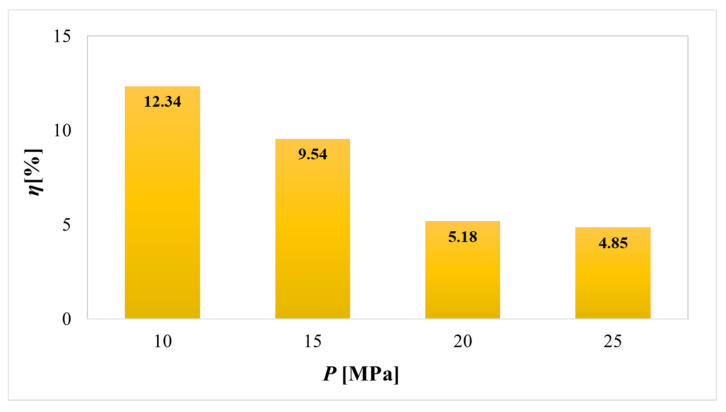
Extraction yields (*η*) obtained by supercritical fluid extraction (SFE) of lyophilized PP using CO_2_ and EtOH as co-solvent at different pressures.

**Figure 4 plants-11-00928-f004:**
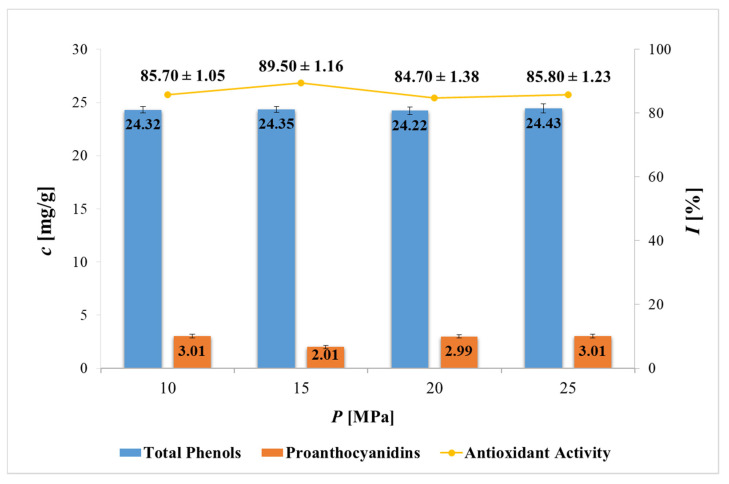
Concentration (c) of total phenols, proanthocyanidins and antioxidant activity (I) of lyophilized PP extracts obtained by supercritical fluid extraction (SFE) using CO_2_ and EtOH as co-solvent at different pressures. Total phenols expressed as mg GAE/g extract. Proanthocyanidins expressed as mg PAC/g extract. Antioxidant activity expressed as % of DPPH inhibition. All experiments were conducted in triplicate and results are reported as a mean ± SD.

**Figure 5 plants-11-00928-f005:**
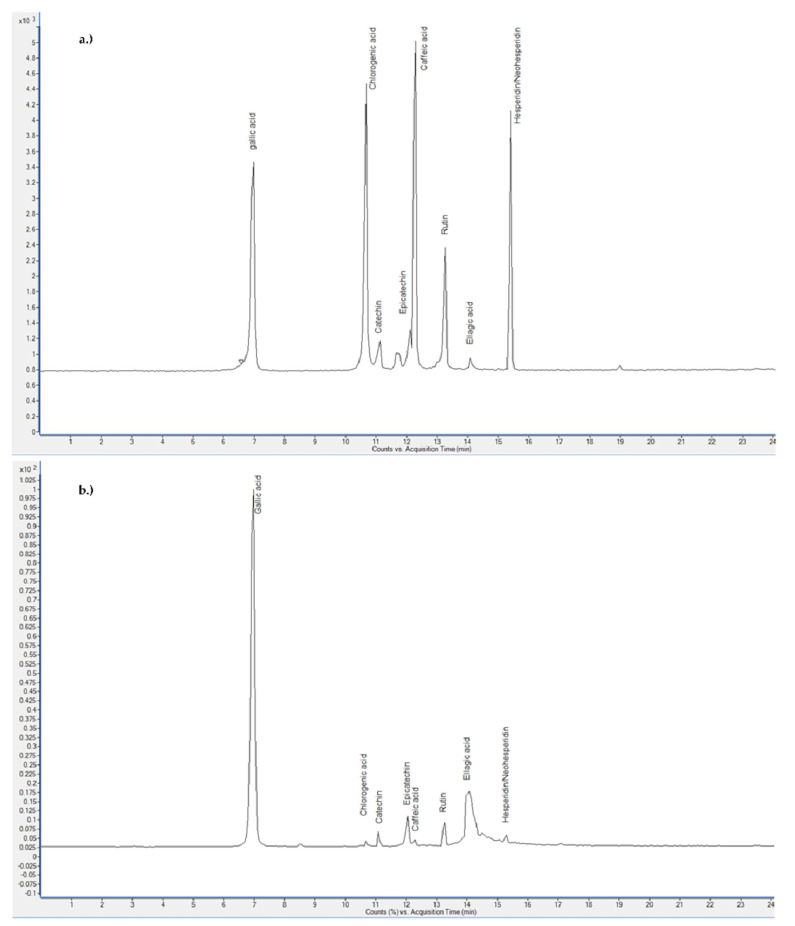
LC-MS/MS chromatograms for (**a**) standards and (**b**) SFE extract obtained at operating pressure of 20 MPa.

**Figure 6 plants-11-00928-f006:**
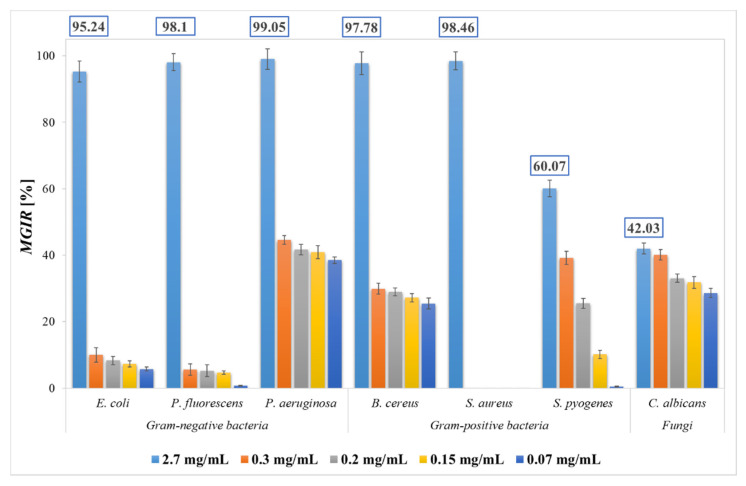
Microbial growth inhibition rates (MGIRs) for lyophilized PP extract obtained by supercritical fluid extraction (SFE) using CO_2_ and EtOH as co-solvent at 20 MPa using 2.7, 0.3, 0.2, 0.15, and 0.07 mg sample/mL microbial suspension. Initial concentrations of bacteria and fungi were 1–5 × 10^6^ CFU/mL.

**Table 1 plants-11-00928-t001:** Total phenols, proanthocyanidins and antioxidant activity of lyophilized PP extracts obtained by ultrasonic extraction (UE), Soxhlet extraction (SE), and cold maceration (CM).

Extraction Method	Solvent	Total Phenols [mg GAE/g]	Proanthocyanidins [mg PAC/g]	Antioxidant Activity [% inhibition]
UE	MeOH	25.92 ± 0.13	3.71 ± 0.08	91.92 ± 1.01
	EtOH	25.41 ± 0.22	2.58 ± 0.04	90.12 ± 2.20
	Ace	24.50 ± 0.11	3.54 ± 0.02	90.50 ± 3.21
SE	MeOH	40.55 ± 0.23	4.47 ± 0.06	93.33 ± 1.55
	EtOH	40.33 ± 0.43	3.26 ± 0.02	91.50 ± 2.18
	Ace	39.26 ± 0.61	2.47 ± 0.03	91.08 ± 1.14
CM	MeOH	42.92 ± 0.65	4.88 ± 0.07	94.35 ± 1.31
	EtOH	40.89 ± 0.81	5.37 ± 0.02	90.97 ± 1.24
	Ace	40.19 ± 0.41	5.82 ± 0.02	91.30 ± 1.52

Total phenols expressed as mg GAE/g extract. Proanthocyanidins expressed as mg PAC/g extract. Antioxidant activity expressed as % of DPPH inhibition. All experiments were conducted in triplicates and results are reported as a mean ± SD.

**Table 2 plants-11-00928-t002:** Polyphenol components in lyophilized PP extracts obtained by ultrasonic extraction (UE), Soxhlet extraction (SE), and cold maceration (CM).

Content (μg/g ± SD)		UE			SE	CM
		MeOH	EtOH	Ace	Ace	Ace
Flavonoids	Catechin	169.37 ± 5.45	78.14 ± 2.69	121.73 ± 2.11	60.84 ± 2.79	246.73 ± 4.83
	Epicatechin	989.13 ± 16.02	350.74 ± 5.55	510.01 ± 14.79	1594.62 ± 69.77	433.64 ± 12.90
	Hesperidin/Neohesperidin	8.85 ± 0.95	22.34 ± 1.05	44.76 ± 1.98	11.20 ± 0.68	41.51 ± 2.11
	Rutin	9.34 ± 0.56	23.65 ± 1.71	54.39 ± 2.44	8.25 ± 1.15	87.02 ± 5.80
Phenolic acids	Caffeic acid	16.25 ± 1.59	12.24 ± 2.55	28.44 ± 1.63	20.33 ± 1.89	18.21 ± 2.00
	Chlorogenic acid	1.33 ± 0.02	2.03 ± 0.03	5.53 ± 0.05	8.41 ± 0.10	2.37 ± 0.02
	Ellagic acid	4585.77 ± 222.12	4798.66 ± 198.25	6883.41 ± 111.89	6377.43 ± 291.11	5601.08 ± 211.87
	Gallic acid	4256.04 ± 187.45	1652.95 ± 52.49	2656.32 ± 105.30	3396.03 ± 99.58	1018.71 ± 20.98
Total content of analyzed polyphenols		10036.08 ± 434.16	6940.74 ± 264.32	10304.60 ± 240.19	11477.11 ± 467.07	7449.26 ± 260.51

Results are expressed as μg of component/g of extract. All experiments were conducted in triplicates and results are reported as a mean ± SD.

**Table 3 plants-11-00928-t003:** Polyphenol components in lyophilized PP extracts obtained by supercritical fluid extraction (SFE) using CO_2_ and EtOH as co-solvent at different pressures and temperature of 40 °C.

Content (μg/g ± SD)		SFE			
		10 MPa	15 MPa	20 MPa	25 MPa
Flavonoids	Catechin	312.39 ± 10.52	366.57 ± 9.85	121.07 ± 6.87	138.33 ± 7.56
	Epicatechin	618.56 ± 20.85	688.27 ± 19.25	718.75 ± 5.85	561.73 ± 11.59
	Hesperidin/Neohesperidin	9.13 ± 0.09	11.71 ± 0.50	62.02 ± 1.36	48.88 ± 2.10
	Rutin	8.90 ± 0.05	55.51 ± 1.23	87.85 ± 2.25	66.91 ± 2.81
Phenolic acids	Caffeic acid	12.29 ± 0.19	10.76 ± 0.20	34.60 ± 0.59	16.41 ± 1.20
	Chlorogenic acid	3.06 ± 0.03	3.61 ± 0.03	12.92 ± 0.05	3.39 ± 0.03
	Ellagic acid	4518.16 ± 157.36	4679.24 ± 209.85	7492.53 ± 345.25	4844.81 ± 221.58
	Gallic acid	1501.49 ± 100.90	1467.13 ± 59.36	3032.11 ± 128.47	1150.95 ± 62.58
Total content of analyzed polyphenols		6983.99 ± 289.99	7282.80 ± 300.27	11561.84 ± 490.69	6831.41 ± 309.45

Results are expressed as μg of component/g of extract. All experiments were conducted in triplicates and results are reported as a mean ± SD.

**Table 4 plants-11-00928-t004:** Qualitative determination of antimicrobial activity of lyophilized PP extract obtained by supercritical fluid extraction (SFE) using CO_2_ and EtOH as co-solvent at 20 MPa. Initial concentrations of bacteria and fungi were 10^5^–10^6^ CFU/mL.

Microorganism		Inhibition Zone [mm]
Gram-negative bacteria	*E. coli*	18 ± 1
	*P. fluorescens*	37 ± 2
	*P. aeruginosa*	31 ± 3
Gram-positive bacteria	*B. cereus*	36 ± 2
	*S. platensis*	-
	*S. aureus*	21 ± 1
	*S. pyogenes*	18 ± 1
Fungi	*A. brasiliensis*	-
	*A. flavus*	11 ± 1
	*A. fumigatus*	-
	*A. niger*	-
	*C. albicans*	18 ± 2
	*P. cyclopium*	11 ± 1
	*S. cerevisiae*	-
	*T. viride*	35 ± 2

**Table 5 plants-11-00928-t005:** Operating conditions for extractions of lyophilized PP.

Extraction Method	Solvent	Temperature [°C]	Solvent Volume [mL]	Time [min]
UE	MeOH	45	150	180
	EtOH			
	Ace			
SE	MeOH	70	150	420
	EtOH	80		360
	Ace	70		300
CM	MeOH	room	150	240
	EtOH			
	Ace			
	**Pressure [MPa]**	**Temperature [°C]**	**Co-Solvent Volume [mL]**	**Time [min]**
SFE	10	40	113	60
	15		101	50
	20		88	45
	25		83	40

**Table 6 plants-11-00928-t006:** Data for analytes identification and quantification using LC-MS/MS method.

Analyte	Rt (min)	Precursor Ion (m/z)	Product Ions (m/z)	CE (V)	Concentration Range (µg/mL)	LOQ (S/N ≥ 10) (µg/mL)
Gallic acid	6.98	169	125, 119	12, 48	0.039–1.960	0.025
Chlorogenic acid	10.63	353	191	24	0.020–1.095	0.020
Catechin	11.11	289	245, 109	4, 16	0.020–0.965	0.020
Epicatechin	12.10	289	245, 109	4, 20	0.021–1.090	0.020
Caffeic acid	12.28	179	135, 107	16, 24	0.250–1.150	0.025
Rutin	13.33	609	300, 271	36, 45	0.019–0.970	0.010
Ellagic acid	14.16	301	145, 229.7	20, 36	0.190–0.965	0.050
Hesperidin/neo	15.40	609	301, 271	35, 40	0.017–0.878	0.010

Rt—Retention Time, CE—Collision Energy, LOQ—Limit of Quantification.

## Data Availability

Not applicable.

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
