# Peer review of "Supercritical Fluid and Conventional Extractions of High Value-Added Compounds from Pomegranate Peels Waste: Production, Quantification and Antimicrobial Activity of Bioactive Constituents"

_plants, 2022, doi:10.3390/plants11070928_

Round 1

Reviewer 1 Report

Kupnik et al. describe the Supercritical fluid and conventional extractions of high value-added compounds from pomegranate peels waste. The phenolic profile in selected extracts was investigated by chromatographic (HPLC) method. Content of different flavonoids and phenolic acids have been determined. SFE extract, obtained at 20 MPa, contained the highest content (11.562 mg/g) of analyzed total polyphenols, with predominant ellagic acid (7.493 24 mg/g). Minimal Inhibitory Concentration (MIC90) was determined as 2.7 mg/mL of SFE pomegranate peel extract in the case of 5 different Gram-negative and Gram-positive bacteria. The artcile is well written and well described and can be accepted after following minor revision.

[1] Importance of fruit peels should be mention in the introduction part

[2] Positive control  is missing for antimicrobial effects

[3] Page 13, lines 406-410: The section "3.2. Preparation of Plant Material for Extractions" should merge with "3.1. Materials" section

Author Response

Reviewer #1:

Kupnik et al. describe the Supercritical fluid and conventional extractions of high value-added compounds from pomegranate peels waste. The phenolic profile in selected extracts was investigated by chromatographic (HPLC) method. Content of different flavonoids and phenolic acids have been determined. SFE extract, obtained at 20 MPa, contained the highest content (11.562 mg/g) of analyzed total polyphenols, with predominant ellagic acid (7.493 24 mg/g). Minimal Inhibitory Concentration (MIC90) was determined as 2.7 mg/mL of SFE pomegranate peel extract in the case of 5 different Gram-negative and Gram-positive bacteria. The artcile is well written and well described and can be accepted after following minor revision.

Answer: We appreciate your time and efforts for reviewing this manuscript. Thank you for all the comments and suggestions we have taken into account. We believe that you remarks were answered precisely and that several improvements have now been included. See below for Answers and Actions.

[1] Importance of fruit peels should be mention in the introduction part

Answer: The importance of fruit peels was mentioned in the introduction part.

Action:

- Page 2, lines 43-47:

, which exhibit antioxidant and antimicrobial effects. Presence of important constituents in fruit peels, which exhibit anti-inflammatory, antioxidant, and antimicrobial effects should be exploited as an opportunity to use these valuable raw materials as a basis for new active substances (e.g., antimicrobials), thus providing an economical and environmentally friendly way to reduce agricultural waste.

[2] Positive control  is missing for antimicrobial effects

Answer: Thank you for this note. Positive controls have been added to the manuscript.

Action:

-Page 14, lines 462-463: For negative control 5% DMSO and deionized H2O were used. As positive controls amoxicillin, nystatin and vancomycin (30μg per disc) were used.

[3] Page 13, lines 406-410: The section "3.2. Preparation of Plant Material for Extractions" should merge with "3.1. Materials" section

Answer: The proposed changes have been made. The following chapters were also renumbered consecutively.

Action:

-Page 12/13, lines 401-406:

…except P. cyclopium and T. viride, which were obtained from Department of Agricultural Chemical Technology, Budapest University of Technology and Economics, Hungary.

3.2. Preparation of Plant Material for Extractions

Lyophilized pomegranate (P. granatum) peels (PP) were purchased from Alfred Galke GmbH (Bad Grund, Germany). Afterwards, the lyophilized peels were used for extractions after grinding.

3.2. Extractions

Reviewer 2 Report

Chromatographic part must be complete.  No confirmation of chemical composition of the extracts was included. Moreover, set of analytes taking into consideration is doubtful. For example, according to literature (doi.org/10.1007/s00216-018-0854-8; doi.org/10.1016/j.chroma.2016.12.017 etc… ) Punica granatum peels contain much more derivatives of ellagic acid then ellagic acid alone. Surprisingly, exactly the same analytes was found by Authors in Red Grape Skin and Rosehip Fruit (previously published paper)

Chromatogram, table with retention times and m/z should be added.

Minor remarks:

The novelty of the study should be emphasized. Authors previously published some investigations carried out using similar scheme: comparison of extraction with different type of solvents using four methods. New plant material is only the novelty of the study?  

The aim of the study should be highlighted.  “Our study is the first to provide a comprehensive insight into the impact of various  PP extractions, including modern SFE.’ – impact on what?

Fig. 1 is unnecessary and could be moved to Supplementary material

Were the results of quantification expressed per g of plant material or per g of dried extracts?

Figure 5: presentation the results as pie chart is not informative. There is doubtful that all polyphenols were  determined.

Some parts of Introduction could be moved to Discussion

Line 386-390: commas are placed incorrectly and therefor this part is misleading.

Author Response

Reviewer #2:

Chromatographic part must be complete.  No confirmation of chemical composition of the extracts was included. Moreover, set of analytes taking into consideration is doubtful. For example, according to literature (doi.org/10.1007/s00216-018-0854-8; doi.org/10.1016/j.chroma.2016.12.017 etc… ) Punica granatum peels contain much more derivatives of ellagic acid then ellagic acid alone. Surprisingly, exactly the same analytes was found by Authors in Red Grape Skin and Rosehip Fruit (previously published paper). Chromatogram, table with retention times and m/z should be added.

Answer: The misleading terms “phenolic profile” was removed from the abstract (line 18-19). “The phenolic compound identification and quantification in selective extracts was done using LC-MS/MS method.”

The HPLC analysis is repeatedly changed to LC-MS/MS analysis throughout the manuscript.

We are aware that the number of components in peel extract is larger than determined in this work. The Punica granatum peel extract contains different groups of components, all of them can not be quantified in one analytical run, the main components were taken into consideration. The manuscript was therefore customized to determination of several components, which according to literature were already identified (J. Agric. Food Chem. 2000, 48, 11, 5331–5337; Journal of Food Composition and Analysis, Vol: 15, Issue: 5, Page: 567-575…). The aim of this work was to quantify the major components together with the minor components in supercritical fluid peel extracts as no literature on these components can be found.

Chromatogram, table with retention times and m/z (Table 6) have been added.

Additionally in Subchapter 2.3, old Figure 5 and Figure 6 have been replaced at the request of the remaining reviewers with Table 2 and Table 3. The LC-MS/MS results units have been changed from mg/g to μg/g in order to add accurate SD values throughout the entire manuscript.

Action:

-Page 1, lines 18-22: The extraction yields, total phenols (TP) and proanthocyanidins (PAC) contents, phenolic profile, and antioxidant activity of different extracts are revealed. TP and PAC recovered by extracts ranged from 24.22 to 42.92 mg gallic acid equivalents (GAE)/g and 2.01 to 5.82 mg PAC/g, respectively. The antioxidant activity of extracts ranged from 84.70 to 94.35%. The phenolic profile in selected extracts was investigated by chromatographic (HPLC) method. The phenolic compound identification and quantification in selective extracts was done using LC-MS/MS method.

-Page 1, line 29: Keywords: antimicrobial activity, antioxidants, bioactive compounds, extraction, HPLC LC-MS/MS, phenolics, phytochemistry, pomegranate, Punica granatum L, secondary metabolites.

-Pages 8-9, lines 251-289:

2.3. Content of certain flavonoids and phenolic acids in PP extracts

Table 2 (UE, SE, CM) and Table 3 (SFE) show the proportions of the selected polyphenol components contents in analyzed PP extracts. The content of 4 flavonoids (catechin, epicatechin, hesperidin/neohesperidin, rutin) and 4 phenolic acids (caffeic acid, chlorogenic acid, ellagic acid, gallic acid) was determined. The common fact in all tables is that ellagic acid (EA) predominated in all analyzed extracts.

When comparing the presence of valuable components in conventional extracts, ellagic acid content was the highest (5601.08-6883.41 μg/g) in Ace extracts (UE > SE > CM), while gallic acid content (the second most abundant component; 1018.71-4256.04 μg/g) was the highest in UE MeOH extract. Caffeic and chlorogenic acid were present in lower concentrations of up to 28.44 and 8.41 μg/g, respectively. Regarding flavonoids, epicatechin (350.74-1594.62 μg/g) was predominant, followed by catechin, rutin and hesperidin/neohesperidin. Overall, the highest total content of analyzed polyphenols was contained in SE Ace extract (11477.11 μg/g), followed by UE Ace, UE MeOH, CM Ace, and UE EtOH extracts of PP.

Table 2. Polyphenol components in lyophilized PP extracts obtained by ultrasonic extraction (UE), Soxhlet extraction (SE), and cold maceration (CM).

Content (μg/g ± SD)

UE

SE

CM

MeOH

EtOH

Ace

Ace

Ace

Flavonoids

Catechin

169.37 ± 5.45

78.14 ± 2.69

121.73 ± 2.11

60.84 ± 2.79

246.73 ± 4.83

Epicatechin

989.13 ± 16.02

350.74 ± 5.55

510.01 ± 14.79

1594.62 ± 69.77

433.64 ± 12.90

Hesperidin/Neohesperidin

8.85 ± 0.95

22.34 ± 1.05

44.76 ± 1.98

11.20 ± 0.68

41.51 ± 2.11

Rutin

9.34 ± 0.56

23.65 ± 1.71

54.39 ± 2.44

8.25 ± 1.15

87.02 ± 5.80

Phenolic acids

Caffeic acid

16.25 ± 1.59

12.24 ± 2.55

28.44 ± 1.63

20.33 ± 1.89

18.21 ± 2.00

Chlorogenic acid

1.33 ± 0.02

2.03 ± 0.03

5.53 ± 0.05

8.41 ± 0.10

2.37 ± 0.02

Ellagic acid

4585.77 ± 222.12

4798.66 ± 198.25

6883.41 ± 111.89

6377.43 ± 291.11

5601.08 ± 211.87

Gallic acid

4256.04 ± 187.45

1652.95 ± 52.49

2656.32 ± 105.30

3396.03 ± 99.58

1018.71 ± 20.98

Total content of analyzed polyphenols

10036.08 ± 434.16

6940.74 ± 264.32

10304.60 ± 240.19

11477.11 ± 467.07

7449.26 ± 260.51

Results are expressed as μg of component/g of extract. All experiments were conducted in triplicates and results are reported as a mean ± SD.

Furthermore, the content of phenolic acids and flavonoids in SFE extracts was completely comparable to conventional extracts. Ellagic acid also predominated in the case of SFE extractions, followed by gallic acid, epicatechin, catechin, rutin, hesperidin/neohesperidin, and caffeic and chlorogenic acid in the lowest concentrations. Here it is necessary to point out mainly SFE extract obtained at operating pressure of 20 MPa (see Figure 5 for chromatogram), in which the highest content (11561.84 μg/g) of analyzed polyphenols was obtained, even higher than in conventional extracts. The same extract contained at least 1.5 times higher concentration (7492.53 μg/g) of ellagic acid than other SFE extracts. The concentrations of the remaining analyzed components were higher in PP extract obtained at 20 MPa, with the exception of catechin, which was present in higher concentrations in SFE extracts obtained at 10, 15, and 25 MPa.

Table 3. Polyphenol components in lyophilized PP extracts obtained by supercritical fluid extraction (SFE) using CO2 and EtOH as co-solvent at different pressures and temperature of 40 °C.

Content (μg/g ± SD)

SFE

10 MPa

15 MPa

20 MPa

25 MPa

Flavonoids

Catechin

312.39 ± 10.52

366.57 ± 9.85

121.07 ± 6.87

138.33 ± 7.56

Epicatechin

618.56 ± 20.85

688.27 ± 19.25

718.75 ± 5.85

561.73 ± 11.59

Hesperidin/Neohesperidin

9.13 ± 0.09

11.71 ± 0.50

62.02 ± 1.36

48.88 ± 2.10

Rutin

8.90 ± 0.05

55.51 ± 1.23

87.85 ± 2.25

66.91 ± 2.81

Phenolic acids

Caffeic acid

12.29 ± 0.19

10.76 ± 0.20

34.60 ± 0.59

16.41 ± 1.20

Chlorogenic acid

3.06 ± 0.03

3.61 ± 0.03

12.92 ± 0.05

3.39 ± 0.03

Ellagic acid

4518.16 ± 157.36

4679.24 ± 209.85

7492.53 ± 345.25

4844.81 ± 221.58

Gallic acid

1501.49 ± 100.90

1467.13 ± 59.36

3032.11 ± 128.47

1150.95 ± 62.58

Total content of analyzed polyphenols

6983.99 ± 289.99

7282.80 ± 300.27

11561.84 ± 490.69

6831.41 ± 309.45

Results are expressed as μg of component/g of extract. All experiments were conducted in triplicates and results are reported as a mean ± SD.

Figure 5. LC-MS/MS chromatograms for a.) standards and b.) SFE extract obtained at operating pressure of 20 MPa.

-Pages 14, lines 447-457:

3.6. Liquid Chromatography-Mass Spectrometry Analysis

To identify and quantify selected phenolic compounds, LC-MS/MS analysis were conducted according to method previously described [39]. For analysis, the Agilent 1200 HPLC was used in tandem with  Agilent 6460 QQQ and JetStream ionization. The analytical column and chromatographic separation conditions were the same as in the aforementioned reference. In brief, the chromatographic column used was Agilent Eclipse Plus, 150 × 4.6 mm i.d., 1.8 μm particle size. Identification of compounds in peel extracts were done using LC-MS/MS method in scan mode using negative polarization and the data for analytes identification and quantification are listed in Table:

Table 6. Data for analytes identification and quantification using LC-MS/MS method

Analyte

Rt (min)

Q1 (m/z)

Q2 (m/z)

CE (V)

Concentration range (µg/mL)

LOQ (S/N 10)

(µg/mL)

Gallic acid

6.98

169

125, 119

12, 48

0.039 – 1.960

0.025

Chlorogenic acid

10.63

353

191

24

0.020 – 1.095

0.020

Catechin

11.11

289

245, 109

4, 16

0.020 – 0.965

0.020

Epicatechin

12.10

289

245, 109

4, 20

0.021 – 1.090

0.020

Caffeic acid

12.28

179

135, 107

16, 24

0.250 – 1.150

0.025

Rutin

13.33

609

300, 271

36, 45

0.019 – 0.970

0.010

Ellagic acid

14.16

301

145, 229.7

20, 36

0.190 – 0.965

0.050

Hesperidin/neo

15.40

609

301, 271

35, 40

0.017 – 0.878

0.010

Minor remarks:

The novelty of the study should be emphasized. Authors previously published some investigations carried out using similar scheme: comparison of extraction with different type of solvents using four methods. New plant material is only the novelty of the study?  

Answer: The novelty of our study is repeatedly emphasized throughout the manuscript. Above all, the comparison and influence of different operating pressures in case of SFE on the yield and content of bioactive components is evaluated as a novelty, and a great contribution is the precise determination of antimicrobial efficacy by determining MGIR for various microorganisms, which in the case of SFE PP extracts have not yet been determined. Moreover, the high content of analyzed polyphenolic components was the highest in case of the PP extract obtained by SFE at 20 MPa (11561.84 μg/g). The high proportion of ellagic acid contributed to the inhibition the growth of Gram-negative, Gram-positive bacteria and fungi.

Action:

-Page 2:

This investigation is focused on bioactive compounds derived from Punica granatum L. peels and highlighted the feasibility of sustainable extraction methods (UE and SFE). It is believed that these findings could be a useful tool for the pomegranate juices industry to apply an effective and economically viable extraction process, transforming a by-product to a high added value functional product with a certain pharmacological effect. Our study is the first to provide a comprehensive insight into the impact of various extraction methods, solvents and process conditions on recovery of high value-added compounds from PP. The importance of our study should be emphasized as it involves the precise determination and comparison of the content of certain polyphenols (e.g., ellagic acid, gallic acid, epicatechin, catechin, and others) in PP extracts obtained with different solvents (methanol (MeOH), ethanol (EtOH), and acetone (Ace)) by conventional extractions and at different operating pressures (10, 15, 20, 25 MPa) by SFE, which has not yet been done.

-Page 3, lines 98-101:

On the other hand, to the best of our knowledge, this is the first extensive study of the antimicrobial activity of SFE PP extracts, which also includes the accurate quantitative determination of the MGIRs at different concentrations of various Gram-negative, Gram-positive bacteria, and fungi.

-Page 15, lines 506-510:

This study provided valuable information regarding different extraction methods of pomegranate peels. Above all, the results of the antimicrobial study of SFE PP extract can help fill the current gap in knowledge in the field of the precise quantitative antimicrobial efficacy of pomegranate peel SFE extracts, which can, based on the current findings, serve as an accessible and extremely potent source for the production of value-added products.

The aim of the study should be highlighted.  “Our study is the first to provide a comprehensive insight into the impact of various PP extractions, including modern SFE.’ – impact on what?

Answer: The purpose and aim of the study is emphasized throughout manuscript, described in detail in the introduction between the lines 79-85 and 96-106. The meaning of the sentence above is now refined and clearer.

Action:

-Page 2, lines 79-81:

Our study is the first to provide a comprehensive insight into the impact of various PP extractions, including modern SFE, on production of high value-added compounds from PP.

Fig. 1 is unnecessary and could be moved to Supplementary material

Answer: The authors believe that the picture is not superfluous and can quickly provide a potential reader with an insight into the chemical structure of important polyphenols in the manuscript.

Were the results of quantification expressed per g of plant material or per g of dried extracts?

Answer: The results of quantifications are expressed per g of extract. All units of the results are explained under Table 1 and also in the subsections where the methods are described.

Figure 5: presentation the results as pie chart is not informative. There is doubtful that all polyphenols were  determined.

Answer: The misleading terms “phenolic profile” was removed from the abstract (line 18-19). “The phenolic compound identification and quantification in selective extracts was done using LC-MS/MS method.” Sentence in line 22-23 was deleted.

We are aware that the number of components in peel extract is larger than determined in this work. The Punica granatum peel extract contains different groups of components, all of them can not be quantified in one analytical run, the main components were taken into consideration. The manuscript was therefore customized to determination of several components, which according to literature were already identified (J. Agric. Food Chem. 2000, 48, 11, 5331–5337; Journal of Food Composition and Analysis, Vol: 15, Issue: 5, Page: 567-575…). The aim of this work was to quantify the major components together with the minor components in supercritical fluid peel extracts as no literature on these components can be found. For better presentation of the results, old Figure 5 and Figure 6 have been replaced with Table 2 and Table 3. The LC-MS/MS results units have been changed from mg/g to μg/g in order to add accurate SD values throughout the entire manuscript.

Additionally, chromatogram and table with retention times and m/z (Table 6) have been added at the request of the remaining reviewers.

Action:

-Page 1, lines 18-22: The extraction yields, total phenols (TP) and proanthocyanidins (PAC) contents, phenolic profile, and antioxidant activity of different extracts are revealed. TP and PAC recovered by extracts ranged from 24.22 to 42.92 mg gallic acid equivalents (GAE)/g and 2.01 to 5.82 mg PAC/g, respectively. The antioxidant activity of extracts ranged from 84.70 to 94.35%. The phenolic profile in selected extracts was investigated by chromatographic (HPLC) method. The phenolic compound identification and quantification in selective extracts was done using LC-MS/MS method.

-Page 1, line 29: Keywords: antimicrobial activity, antioxidants, bioactive compounds, extraction, HPLC LC-MS/MS, phenolics, phytochemistry, pomegranate, Punica granatum L, secondary metabolites.

-Pages 8-9, lines 251-289:

2.3. Content of certain flavonoids and phenolic acids in PP extracts

Table 2 (UE, SE, CM) and Table 3 (SFE) show the proportions of the selected polyphenol components contents in analyzed PP extracts. The content of 4 flavonoids (catechin, epicatechin, hesperidin/neohesperidin, rutin) and 4 phenolic acids (caffeic acid, chlorogenic acid, ellagic acid, gallic acid) was determined. The common fact in all tables is that ellagic acid (EA) predominated in all analyzed extracts.

When comparing the presence of valuable components in conventional extracts, ellagic acid content was the highest (5601.08-6883.41 μg/g) in Ace extracts (UE > SE > CM), while gallic acid content (the second most abundant component; 1018.71-4256.04 μg/g) was the highest in UE MeOH extract. Caffeic and chlorogenic acid were present in lower concentrations of up to 28.44 and 8.41 μg/g, respectively. Regarding flavonoids, epicatechin (350.74-1594.62 μg/g) was predominant, followed by catechin, rutin and hesperidin/neohesperidin. Overall, the highest total content of analyzed polyphenols was contained in SE Ace extract (11477.11 μg/g), followed by UE Ace, UE MeOH, CM Ace, and UE EtOH extracts of PP.

Table 2. Polyphenol components in lyophilized PP extracts obtained by ultrasonic extraction (UE), Soxhlet extraction (SE), and cold maceration (CM).

Content (μg/g ± SD)

UE

SE

CM

MeOH

EtOH

Ace

Ace

Ace

Flavonoids

Catechin

169.37 ± 5.45

78.14 ± 2.69

121.73 ± 2.11

60.84 ± 2.79

246.73 ± 4.83

Epicatechin

989.13 ± 16.02

350.74 ± 5.55

510.01 ± 14.79

1594.62 ± 69.77

433.64 ± 12.90

Hesperidin/Neohesperidin

8.85 ± 0.95

22.34 ± 1.05

44.76 ± 1.98

11.20 ± 0.68

41.51 ± 2.11

Rutin

9.34 ± 0.56

23.65 ± 1.71

54.39 ± 2.44

8.25 ± 1.15

87.02 ± 5.80

Phenolic acids

Caffeic acid

16.25 ± 1.59

12.24 ± 2.55

28.44 ± 1.63

20.33 ± 1.89

18.21 ± 2.00

Chlorogenic acid

1.33 ± 0.02

2.03 ± 0.03

5.53 ± 0.05

8.41 ± 0.10

2.37 ± 0.02

Ellagic acid

4585.77 ± 222.12

4798.66 ± 198.25

6883.41 ± 111.89

6377.43 ± 291.11

5601.08 ± 211.87

Gallic acid

4256.04 ± 187.45

1652.95 ± 52.49

2656.32 ± 105.30

3396.03 ± 99.58

1018.71 ± 20.98

Total content of analyzed polyphenols

10036.08 ± 434.16

6940.74 ± 264.32

10304.60 ± 240.19

11477.11 ± 467.07

7449.26 ± 260.51

Results are expressed as μg of component/g of extract. All experiments were conducted in triplicates and results are reported as a mean ± SD.

Furthermore, the content of phenolic acids and flavonoids in SFE extracts was completely comparable to conventional extracts. Ellagic acid also predominated in the case of SFE extractions, followed by gallic acid, epicatechin, catechin, rutin, hesperidin/neohesperidin, and caffeic and chlorogenic acid in the lowest concentrations. Here it is necessary to point out mainly SFE extract obtained at operating pressure of 20 MPa (see Figure 5 for chromatogram), in which the highest content (11561.84 μg/g) of analyzed polyphenols was obtained, even higher than in conventional extracts. The same extract contained at least 1.5 times higher concentration (7492.53 μg/g) of ellagic acid than other SFE extracts. The concentrations of the remaining analyzed components were higher in PP extract obtained at 20 MPa, with the exception of catechin, which was present in higher concentrations in SFE extracts obtained at 10, 15, and 25 MPa.

Table 3. Polyphenol components in lyophilized PP extracts obtained by supercritical fluid extraction (SFE) using CO2 and EtOH as co-solvent at different pressures and temperature of 40 °C.

Content (μg/g ± SD)

SFE

10 MPa

15 MPa

20 MPa

25 MPa

Flavonoids

Catechin

312.39 ± 10.52

366.57 ± 9.85

121.07 ± 6.87

138.33 ± 7.56

Epicatechin

618.56 ± 20.85

688.27 ± 19.25

718.75 ± 5.85

561.73 ± 11.59

Hesperidin/Neohesperidin

9.13 ± 0.09

11.71 ± 0.50

62.02 ± 1.36

48.88 ± 2.10

Rutin

8.90 ± 0.05

55.51 ± 1.23

87.85 ± 2.25

66.91 ± 2.81

Phenolic acids

Caffeic acid

12.29 ± 0.19

10.76 ± 0.20

34.60 ± 0.59

16.41 ± 1.20

Chlorogenic acid

3.06 ± 0.03

3.61 ± 0.03

12.92 ± 0.05

3.39 ± 0.03

Ellagic acid

4518.16 ± 157.36

4679.24 ± 209.85

7492.53 ± 345.25

4844.81 ± 221.58

Gallic acid

1501.49 ± 100.90

1467.13 ± 59.36

3032.11 ± 128.47

1150.95 ± 62.58

Total content of analyzed polyphenols

6983.99 ± 289.99

7282.80 ± 300.27

11561.84 ± 490.69

6831.41 ± 309.45

Results are expressed as μg of component/g of extract. All experiments were conducted in triplicates and results are reported as a mean ± SD.

Figure 5. LC-MS/MS chromatograms for a.) standards and b.) SFE extract obtained at operating pressure of 20 MPa.

-Pages 14, lines 447-457:

3.6. Liquid Chromatography-Mass Spectrometry Analysis

To identify and quantify selected phenolic compounds, LC-MS/MS analysis were conducted according to method previously described [39]. For analysis, the Agilent 1200 HPLC was used in tandem with  Agilent 6460 QQQ and JetStream ionization. The analytical column and chromatographic separation conditions were the same as in the aforementioned reference. In brief, the chromatographic column used was Agilent Eclipse Plus, 150 × 4.6 mm i.d., 1.8 μm particle size. Identification of compounds in peel extracts were done using LC-MS/MS method in scan mode using negative polarization and the data for analytes identification and quantification are listed in Table 6:

Table 6. Data for analytes identification and quantification using LC-MS/MS method

Analyte

Rt (min)

Q1 (m/z)

Q2 (m/z)

CE (V)

Concentration range (µg/mL)

LOQ (S/N 10)

(µg/mL)

Gallic acid

6.98

169

125, 119

12, 48

0.039 – 1.960

0.025

Chlorogenic acid

10.63

353

191

24

0.020 – 1.095

0.020

Catechin

11.11

289

245, 109

4, 16

0.020 – 0.965

0.020

Epicatechin

12.10

289

245, 109

4, 20

0.021 – 1.090

0.020

Caffeic acid

12.28

179

135, 107

16, 24

0.250 – 1.150

0.025

Rutin

13.33

609

300, 271

36, 45

0.019 – 0.970

0.010

Ellagic acid

14.16

301

145, 229.7

20, 36

0.190 – 0.965

0.050

Hesperidin/neo

15.40

609

301, 271

35, 40

0.017 – 0.878

0.010

Some parts of Introduction could be moved to Discussion

Answer: The suggestion was accepted, minor changes were made. See Action below.

Action:

-From Page 2, line 75 (text marked red) to page 5, lines 144-151:

According to the reviewed literature, a small number of studies have reported on extraction of bioactive components from PP using SFE. Mushtaq et al. [16] used enzyme-assisted SFE for extraction of phenolic antioxidants from PP, while Bustamante et al. [17] and Rivas et al. [18] used supercritical CO2 using a Box-Behnken design, where temperature, pressure and co-solvent or time were independent variables for optimization of process. Studies revealed that the highest yields were obtained after 2.5h extraction at 40-50 °C and 20-30 MPa using 20% of co-solvent. Recently Rivas et al. [18] reported 1-1.5% yield for SFE of PP, using SC CO2 at 25-30 MPa and 45-55 °C, while Ara and Raofie [19] achieved 1.18% yield using SC CO2 and MeOH as modifier at 35 MPa and 55 °C.

Line 386-390: commas are placed incorrectly and therefor this part is misleading.

Answer: Thank you for the note, changes have been made to make this part more understandable as well.

Action:

-Page 12, lines 384-388:

Acetonitrile, agar, 2,2-diphenyl-1-picrylhydrazyl (DPPH, ≥97.0%), hydrochloric acid (HCl,37%), dimethyl sulfoxide (DMSO), ethanol (EtOH, ≥99.5%), ferrous sulfate heptahydrate (Fe(SO4)×7H2O), Folin-Ciocalteu’s reagent (FC), gallic acid (GA, 97.5-102.5%), methanol (MeOH), n-butanol (≥99.5%), sodium carbonate (Na2CO3, ≥99.9%), peptone from soybean and yeast extract, were purchased from Sigma Aldrich® (Darmstadt, Germany).

Reviewer 3 Report

The study by Kaja Kupnik et al entitled “Supercritical fluid and conventional extractions of high value added compounds from pomegranate peels waste: production, characterization and quantification of bioactive constituents” has been reviewed. The study provides a comprehensive insight into the impact of various pomegranate peels extractions, including supercritical fluid extraction together with the antimicrobial activity of extracts from supercritical fluid extraction. The manuscript is potentially interesting, the rationale is clear, figures and legends are exhaustive and the flow is well structured. Some minor suggestions are listed below.

Please include in the title the antimicrobial activities.

Please reduce the introduction, in some parts, it results redundant (lines 1-46 and lines 97-114). Don't list microorganisms species.

Considering the type of scientific work presented, it is essential to describe in detail the extraction methods used. Reference 37 does not contain sufficient information for the correct reproducibility of the extracts. Please report the extraction methods in detail including extraction time.

Please review the conclusions by making them shorter.

I would urge the authors to accommodate the suggestions reported above, to further improve the quality of the manuscript.

Author Response

Reviewer #3:

The study by Kaja Kupnik et al entitled “Supercritical fluid and conventional extractions of high value added compounds from pomegranate peels waste: production, characterization and quantification of bioactive constituents” has been reviewed. The study provides a comprehensive insight into the impact of various pomegranate peels extractions, including supercritical fluid extraction together with the antimicrobial activity of extracts from supercritical fluid extraction. The manuscript is potentially interesting, the rationale is clear, figures and legends are exhaustive and the flow is well structured. Some minor suggestions are listed below.

Answer: We appreciate your time and efforts for reviewing this manuscript. See below for Answers and Actions that include all changes made based on your suggestions.

Please include in the title the antimicrobial activities.

Answer: Title of the manuscript has been slightly changed to comply your request.

Action: Supercritical fluid and conventional extractions of high value-added compounds from pomegranate peels waste: production, characterization and quantification and antimicrobial activity of bioactive constituents

Please reduce the introduction, in some parts, it results redundant (lines 1-46 and lines 97-114). Don't list microorganisms species.

Answer: Introduction has been reduced. Microorganism species are not listed anymore.

Action:

-Page 1, line 39:

Increasing interest into prevention, reuse and recover of food by-products is since various fruit by-products represent a rich source of valuable bioactive compounds that can be obtained through different biotechnological methodologies.

-From Page 2, line 75 (text marked red) was moved to page 5, lines 144-151:

According to the reviewed literature, a small number of studies have reported on extraction of bioactive components from PP using SFE. Mushtaq et al. [16] used enzyme-assisted SFE for extraction of phenolic antioxidants from PP, while Bustamante et al. [17] and Rivas et al. [18] used supercritical CO2 using a Box-Behnken design, where temperature, pressure and co-solvent or time were independent variables for optimization of process. Studies revealed that the highest yields were obtained after 2.5h extraction at 40-50 °C and 20-30 MPa using 20% of co-solvent. Recently Rivas et al. [18] reported 1-1.5% yield for SFE of PP, using SC CO2 at 25-30 MPa and 45-55 °C, while Ara and Raofie [19] achieved 1.18% yield using SC CO2 and MeOH as modifier at 35 MPa and 55 °C.

-Page 3, lines 101-106:

Antimicrobial activity was qualitatively determined against 3 species of Gram-negative bacteria (Escherichia coli, Pseudomonas aeruginosa, Pseudomonas fluorescens), 4 species of Gram-positive bacteria (B. cereus, S. aureus, Streptococcus pyogenes, Streptomyces platensis) and 8 species of fungi (Aspergillus brasiliensis, Aspergillus flavus, Aspergillus fumigatus, Aspergillus niger, Candida albicans, Penicillium cyclopium, Saccharomyces cerevisiae, Trichoderma viride). Furthermore, the antimicrobial efficacy of PP extract obtained by SFE was quantified by determination of MGIR at 5 different concentrations and MIC90 values for PP extract against 7 species (E. coli, P. aeruginosa, P. fluorescens, B. cereus, S. aureus, S. pyogenes, and C. albicans) of microorganisms.

Considering the type of scientific work presented, it is essential to describe in detail the extraction methods used. Reference 37 does not contain sufficient information for the correct reproducibility of the extracts. Please report the extraction methods in detail including extraction time.

Answer: The reference discloses the process of used extractions. Now additional sufficient information on the operating conditions of all extractions has been added in Table 3, including temperature, solvent volume, and extraction time in the case of conventional extractions and temperature, co-solvent volume and extraction time in the case of SFE.

Action:

-Page 14, lines 421-422:

Table 3. Operating conditions for extractions of lyophilized PP

Extraction Method

Solvent

Temperature

[°C]

Solvent volume

[mL]

Time

[min]

UE

MeOH

45

150

180

EtOH

Ace

SE

MeOH

70

150

420

EtOH

80

360

Ace

70

300

CM

MeOH

room

150

240

EtOH

Ace

Pressure

[MPa]

Temperature

[°C]

Co-solvent volume

[mL]

Time

[min]

SFE

10

40

113

60

15

101

50

20

88

45

25

83

40

Please review the conclusions by making them shorter.

Answer: Some general facts have been deleted from and the chapter Conclusions is now a bit shorter. We believe that the remaining information is important conclusions and facts that must remain stated so that the potential reader can quickly learn important findings from many results.

Action:

-Page 15, line 485:

Pomegranate peels, a by-product in the manufacture of pomegranate juice and other products used mainly in the food industry, have been investigated as a source of high value-added compounds.

I would urge the authors to accommodate the suggestions reported above, to further improve the quality of the manuscript.

Answer: Thank you for all the comments and suggestions we have taken into account. We believe that you remarks were answered precisely and that several improvements have now been included.

Reviewer 4 Report

Dear Authors,

Please find my suggestions within the pdf file of the manuscript. I recommend the publication after taking into account my comments.

Best regards,

Reviewer

Author Response

Reviewer #4:

Dear Authors,

Please find my suggestions within the pdf file of the manuscript. I recommend the publication after taking into account my comments. plants-1645571-review.pdf

Best regards,

Reviewer

Answer: We appreciate your time and efforts for reviewing this manuscript. Thank you for all the comments and suggestions we have taken into account. We believe that you remarks were answered precisely and that several improvements have now been included. See below for Actions.

Actions:

-Page 1, lines 25-28 (from zero to ten the numbers should be written!):

For the first time, Microbial Growth Inhibition Rates (MGIRs) were determined at 5 five different concentrations of pomegranate SFE extract against 7 seven microorganisms. Minimal Inhibitory Concentration (MIC90) was determined as 2.7 mg/mL of SFE pomegranate peel extract in the case of 5 five different Gram-negative and Gram-positive bacteria.

-Page 5, lines 175-176 (f!):

…, followed by EtOH and Ace, which coincides with the polarity odf the solvents (MeOH > EtOH > Ace) and literature [20,21].

- Page 6, lines 177-180 (Please mention if this is SD!):

Table 1. Total phenols, proanthocyanidins and antioxidant activity of lyophilized PP extracts obtained by ultrasonic extraction (UE), Soxhlet extraction (SE), and cold maceration (CM)

Extraction

Method

Solvent

Total Phenols

[mg GAE/g]

Proanthocyanidins

[mg PAC/g]

Antioxidant Activity

[% inhibition]

UE

MeOH

25.92 ± 0.13

3.71 ± 0.08

91.92 ± 1.01

EtOH

25.41 ± 0.22

2.58 ± 0.04

90.12 ± 2.20

Ace

24.50 ± 0.11

3.54 ± 0.02

90.50 ± 3.21

SE

MeOH

40.55 ± 0.23

4.47 ± 0.06

93.33 ± 1.55

EtOH

40.33 ± 0.43

3.26 ± 0.02

91.50 ± 2.18

Ace

39.26 ± 0.61

2.47 ± 0.03

91.08 ± 1.14

CM

MeOH

42.92 ± 0.65

4.88 ± 0.07

94.35 ± 1.31

EtOH

40.89 ± 0.81

5.37 ± 0.02

90.97 ± 1.24

Ace

40.19 ± 0.41

5.82 ± 0.02

91.30 ± 1.52

Total phenols expressed as mg GAE/g extract. Proanthocyanidins expressed as mg PAC/g extract. Antioxidant activity expressed as % of DPPH inhibition. All experiments were conducted in triplicates and results are reported as a mean ± SD.

-Page 7, lines 214-219 (Please, put the SD to each column!):

Figure 4. Concentration (c) of total phenols, proanthocyanidins and antioxidant activity (I) of lyophilized PP extracts obtained by supercritical fluid extraction (SFE) using CO2 and EtOH as co-solvent at different pressures. Total phenols expressed as mg GAE/g extract. Proanthocyanidins expressed as mg PAC/g extract. Antioxidant activity expressed as % of DPPH inhibition. All experiments were conducted in triplicates and results are reported as a mean ± SD.

-Page 8/9, old Figure 5 and Figure 6 (Please, put the SD to each column!): For better presentation of the results, old Figure 5 and Figure 6 have been replaced with Table 2 and Table 3. The LC-MS/MS results units have been changed from mg/g to μg/g in order to add accurate SD values.

Table 2. Polyphenol components in lyophilized PP extracts obtained by ultrasonic extraction (UE), Soxhlet extraction (SE), and cold maceration (CM).

Content (μg/g ± SD)

UE

SE

CM

MeOH

EtOH

Ace

Ace

Ace

Flavonoids

Catechin

169.37 ± 5.45

78.14 ± 2.69

121.73 ± 2.11

60.84 ± 2.79

246.73 ± 4.83

Epicatechin

989.13 ± 16.02

350.74 ± 5.55

510.01 ± 14.79

1594.62 ± 69.77

433.64 ± 12.90

Hesperidin/Neohesperidin

8.85 ± 0.95

22.34 ± 1.05

44.76 ± 1.98

11.20 ± 0.68

41.51 ± 2.11

Rutin

9.34 ± 0.56

23.65 ± 1.71

54.39 ± 2.44

8.25 ± 1.15

87.02 ± 5.80

Phenolic acids

Caffeic acid

16.25 ± 1.59

12.24 ± 2.55

28.44 ± 1.63

20.33 ± 1.89

18.21 ± 2.00

Chlorogenic acid

1.33 ± 0.02

2.03 ± 0.03

5.53 ± 0.05

8.41 ± 0.10

2.37 ± 0.02

Ellagic acid

4585.77 ± 222.12

4798.66 ± 198.25

6883.41 ± 111.89

6377.43 ± 291.11

5601.08 ± 211.87

Gallic acid

4256.04 ± 187.45

1652.95 ± 52.49

2656.32 ± 105.30

3396.03 ± 99.58

1018.71 ± 20.98

Total content of analyzed polyphenols

10036.08 ± 434.16

6940.74 ± 264.32

10304.60 ± 240.19

11477.11 ± 467.07

7449.26 ± 260.51

Results are expressed as μg of component/g of extract. All experiments were conducted in triplicates and results are reported as a mean ± SD.

Table 3. Polyphenol components in lyophilized PP extracts obtained by supercritical fluid extraction (SFE) using CO2 and EtOH as co-solvent at different pressures and temperature of 40 °C.

Content (μg/g ± SD)

SFE

10 MPa

15 MPa

20 MPa

25 MPa

Flavonoids

Catechin

312.39 ± 10.52

366.57 ± 9.85

121.07 ± 6.87

138.33 ± 7.56

Epicatechin

618.56 ± 20.85

688.27 ± 19.25

718.75 ± 5.85

561.73 ± 11.59

Hesperidin/Neohesperidin

9.13 ± 0.09

11.71 ± 0.50

62.02 ± 1.36

48.88 ± 2.10

Rutin

8.90 ± 0.05

55.51 ± 1.23

87.85 ± 2.25

66.91 ± 2.81

Phenolic acids

Caffeic acid

12.29 ± 0.19

10.76 ± 0.20

34.60 ± 0.59

16.41 ± 1.20

Chlorogenic acid

3.06 ± 0.03

3.61 ± 0.03

12.92 ± 0.05

3.39 ± 0.03

Ellagic acid

4518.16 ± 157.36

4679.24 ± 209.85

7492.53 ± 345.25

4844.81 ± 221.58

Gallic acid

1501.49 ± 100.90

1467.13 ± 59.36

3032.11 ± 128.47

1150.95 ± 62.58

Total content of analyzed polyphenols

6983.99 ± 289.99

7282.80 ± 300.27

11561.84 ± 490.69

6831.41 ± 309.45

Results are expressed as μg of component/g of extract. All experiments were conducted in triplicates and results are reported as a mean ± SD.

-Page 11, Figure 6 (Could you please enlarge a little bit the graphic area to see better the SD of the highest blue columns?):

Figure 6. Microbial growth inhibition rates (MGIRs) for lyophilized PP extract obtained by supercritical fluid extraction (SFE) using CO2 and EtOH as co-solvent at 20 MPa using 2.7, 0.3, 0.2, 0.15, and 0.07 mg sample/mL microbial suspension. Initial concentrations of bacteria and fungi were 1–5 × 106 CFU/mL.

- Page 14, line 447 (Please, write the entire name of the method! LC-MS, because this method you mention to use!):

3.6. HPLC Liquid Chromatography-Mass Spectrometry Analysis

Round 2

Reviewer 2 Report

I conditionally accept Authors' explanations, but in future, they should provide better documentation of the results (MS spectra); especially that they identified in plant material compounds  not found by other researchers (such as hesperidin). Authors as justification cited two papers (J. Agric. Food Chem. 2000, 48, 11, 5331–5337; Journal of Food Composition and Analysis, Vol: 15, Issue: 5, Page: 567-575…) but in the manuscripts identification was not  based on MS analysis and set of identified analytes is not the same as in Authors’ study. Moreover, it is extremely  unusual (and questionable) that the same analytes  were found by Authors in Red Grape Skin and Rosehip Fruit

Minor comment: In Table 6 should be m/z-H. Abbreviations:  Q1, Q2, CE should be explained under the table.

Author Response

Dear Reviewer,

herewith is our revised manuscript entitled "Supercritical fluid and conventional extractions of high value-added compounds from pomegranate peels waste: production, quantification and antimicrobial activity of bioactive constituents", authors Kaja Kupnik, Maja Leitgeb, Mateja Primožič, Vesna Postružnik, Petra Kotnik, Nika Kučuk, Željko Knez, and Maša Knez Marevci (Manuscript ID: plants-1645571).

The manuscript was revised by the authors and all required changes were made.

Thank you for your time and effort. Please address all correspondence to:

Reviewer #2:

I conditionally accept Authors' explanations, but in future, they should provide better documentation of the results (MS spectra); especially that they identified in plant material compounds  not found by other researchers (such as hesperidin). Authors as justification cited two papers (J. Agric. Food Chem. 2000, 48, 11, 5331–5337; Journal of Food Composition and Analysis, Vol: 15, Issue: 5, Page: 567-575…) but in the manuscripts identification was not  based on MS analysis and set of identified analytes is not the same as in Authors’ study. Moreover, it is extremely  unusual (and questionable) that the same analytes  were found by Authors in Red Grape Skin and Rosehip Fruit

Answer: Thank you for your comment. Like mentioned in first round of revision, the Punica granatum peel extract contains different groups of components, the main components were taken into consideration. The manuscript was therefore customized to determination of several components, which according to literature were already identified and it was mentioned in the manuscript on page 2, lines 57-62. As for the remaining compounds, hesperidin has already been mentioned and identified in Punica granatum peel extracts according to literature (Heliyon, vol. 5 (4), 2019, e01575; J. Food Biochem. 2019; 43: 1–9, Molecules. 2020 Jun; 25 (12) : 2859). Additionally, reference [6] Gullón et. al. (Moleciles 2020) has been added to the manuscript to support these findings. In our study, all components were identified prior to quantification and, if not present in Punica granatum peels, could not be quantified due to the mass spectrometry selectivity.

Action:

-Page 2, lines 57-62: However, it should be noted that pomegranate peels (PP) are considered a rich source of polyphenols. Polyphenols present in PP include anthocyanins (i.e., cyanidin, delphinidin, pelargonidin), flavonoids (i.e., catechin, epicatechin, quercetin, rutin, kaempferol, luteolin, naringenin, hesperidin), phenolic acids (i.e., caffeic, chlorogenic, ellagic acid, and gallic acid), and ellagitannins (i.e., punicalagin and punicalin) [5,6].

Minor comment: In Table 6 should be m/z-H. Abbreviations:  Q1, Q2, CE should be explained under the table.

Answer: Table 6 has been supplemented with the necessary information.

Action:

-Page 14, lines 461-462:

Table 6. Data for analytes identification and quantification using LC-MS/MS method

Analyte

Rt (min)

Q1 Precursor ion (m/z)

Q2 Product ions (m/z)

CE (V)

Concentration range (µg/mL)

LOQ (S/N 10)

(µg/mL)

Gallic acid

6.98

169

125, 119

12, 48

0.039 – 1.960

0.025

Chlorogenic acid

10.63

353

191

24

0.020 – 1.095

0.020

Catechin

11.11

289

245, 109

4, 16

0.020 – 0.965

0.020

Epicatechin

12.10

289

245, 109

4, 20

0.021 – 1.090

0.020

Caffeic acid

12.28

179

135, 107

16, 24

0.250 – 1.150

0.025

Rutin

13.33

609

300, 271

36, 45

0.019 – 0.970

0.010

Ellagic acid

14.16

301

145, 229.7

20, 36

0.190 – 0.965

0.050

Hesperidin/neo

15.40

609

301, 271

35, 40

0.017 – 0.878

0.010

Rt- Retention Time, CE – Collision Energy, LOQ - Limit of Quantification